# Poincaré maps for analyzing complex hierarchies in single-cell data

Anna Klimovskaia [1✉], David Lopez-Paz[1], Léon Bottou [2] & Maximilian Nickel[2✉]

The need to understand cell developmental processes spawned a plethora of computational methods for discovering hierarchies from scRNAseq data. However, existing techniques are based on Euclidean geometry, a suboptimal choice for modeling complex cell trajectories with multiple branches. To overcome this fundamental representation issue we propose Poincaré maps, a method that harness the power of hyperbolic geometry into the realm of single-cell data analysis. Often understood as a continuous extension of trees, hyperbolic geometry enables the embedding of complex hierarchical data in only two dimensions while preserving the pairwise distances between points in the hierarchy. This enables the use of our embeddings in a wide variety of downstream data analysis tasks, such as visualization, clustering, lineage detection and pseudotime inference. When compared to existing methods — unable to address all these important tasks using a single embedding — Poincaré maps produce state-of-the-art two-dimensional representations of cell trajectories on multiple scRNAseq datasets.

[1] Facebook AI, 6 Rue Ménars, Paris 75002, France. [2] Facebook AI, 770 Broadway, New York, NY 10003, USA. ✉email: klanna@fb.com; maxn@fb.com

Understanding cellular differentiation, e.g., the transition of immature cells into specialized types, is a central task in modern developmental biology. Recent advances in single-cell technologies, such as single-cell RNA sequencing and mass cytometry, enabled important insights into these processes based on high-throughput cell measurements[1–4]. Computational methods to accurately discover and represent cell development processes from large datasets and noisy measurements are therefore in great demand. This is a challenging task since methods are required to reveal the progression of cells along continuous trajectories with tree-like structures and multiple branches (e.g., as in Waddington's classic epigenetic landscape[5]). Multiple advances have been made towards this goal of discovering and analyzing hierarchical structures from single-cell measurements[6]. In particular, methods leveraging assumptions about the hierarchical structure for visualization[7–12], clustering[13,14], and pseudotime inference[15,16], have fueled successful applications of data analysis in developmental biology. To visualize hierarchical relationships in cell development, many state-of-the-art methods embed cell measurements in low-dimensional Euclidean spaces[7,8,17,18]. However, this approach is limited when modeling complex hierarchies, as low-dimensional Euclidean embeddings distort pairwise distances between measurements substantially. The resulting embeddings are problematic not only for visualization but also for other downstream tasks such as clustering and lineage identification.

To overcome the issues of data dimensionality reduction in Euclidean spaces, we propose Poincaré maps, a method to compute embeddings in hyperbolic spaces. These enable multiple advantages. First, hyperbolic spaces can be thought of as a continuous analog to trees and enable low-distortion embeddings of hierarchical structures in as few as two dimensions[19]. Second, the metric structure of hyperbolic spaces retains the ability to model continuous trajectories using pairwise distances of measurements, and allows us to employ the obtained embeddings in downstream tasks such as clustering, lineage detection, and pseudotime inference. Third, the Riemannian structure of hyperbolic

manifolds enables the use of gradient-based optimization methods what is essential to compute embeddings of large-scale measurements. Fourth, while we follow Nickel and Kiela[20] to leverage the Poincaré disk as an embedding space, we are first to employ pairwise distances obtained from a nearest-neighbor graph as a learning signal to construct hyperbolic embeddings for the discovery of complex hierarchies in data.

An important property of Poincaré maps is that it allows us to approach all these different tasks using a single embedding, by combining the identification of clusters, trajectories, and hierarchies in an unsupervised manner. To the best of our knowledge, this is not possible with existing methods, which we review in the following. t-SNE[17] is a state-of-the-art visualization method that exploits local similarities to achieve visual separation of the clusters in the data. However, t-SNE does not preserve global similarities between clusters and therefore does not guarantee that the global hierarchical structure will be preserved. UMAP[18] computes a low-dimensional Euclidean representation of data that preserves the topological structure. However, there are no guarantees that there exists a low-dimensional representation of complex tree topologies in a two-dimensional Euclidean space. Diffusion maps[7] specifically tackles the problem of capturing diffusion-like dynamics and continuous branching in the data. However, it allows us to visualize only simple branching structures in two dimensions. Graph abstractions[8] (PAGA) and Monocle 2[15] are another class of methods to capture and visualize hierarchical relationships in the data. PAGA produces an "abstracted graph" with nodes corresponding to partitions of the data, and edges representing relationships between these nodes. PAGA does not represent the relationships within partitions. However, PAGA can be used to initialize UMAP and ForceAtlas2, as done by the authors. Despite the fact that ForceAtlas2[21] produces a good visual layout of tree topology, it does not preserve hierarchical distances. PHATE[22], a method that has been demonstrated able to recover hierarchies with multiple branches, is also affected by the distortion artifacts of Euclidean spaces. Monocle 2[15] forces

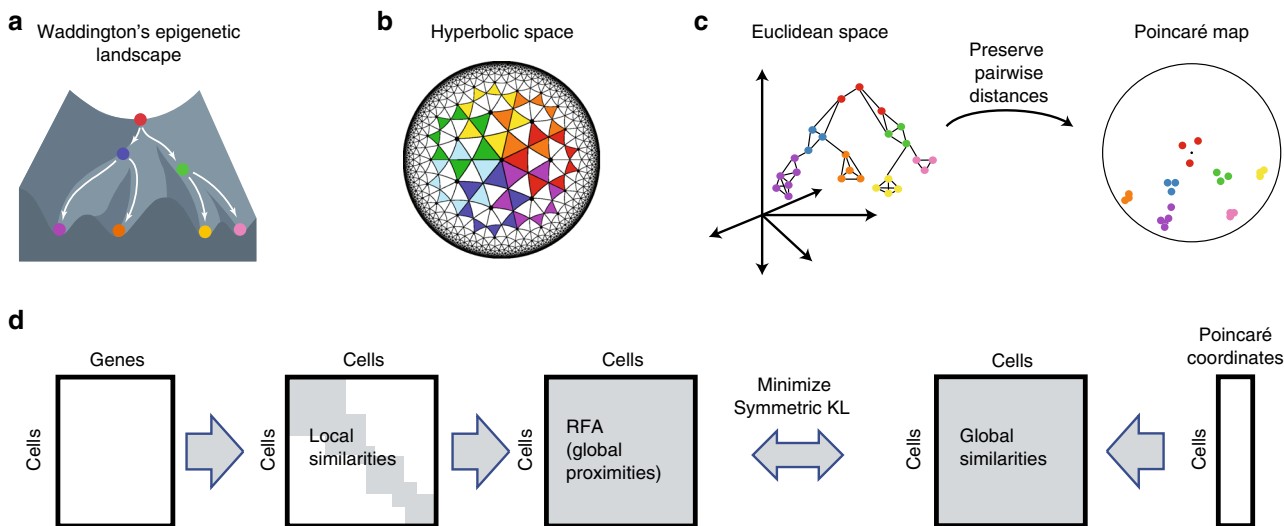

**Fig. 1 Poincaré maps discover hierarchies and branching processes. a** Our goal is to recover cell developmental processes, depicted here on the Waddington's epigenetic landscape. **b** Poincaré disks provide a natural geometry to preserve hierarchical structures and pairwise similarities in two dimensions. Poincaré disks grow as we approach their boundary: all the triangles depicted in the figure are of equal size. **c** Poincaré maps first estimate geodesic distances, computed from a connected *k*-nearest-neighbor graph. Second, they compute two-dimensional hyperbolic embeddings that preserve these similarities. **d** Overview of Poincaré maps embedding procedure. From a given feature matrix, Poincaré maps firsts estimates local similarities based on a user specified local distance metric (Euclidean, cosine, etc.) and Gaussian kernel with a tunable parameter $\sigma$. Local similarities then used to compute global proximities on the dataset. By means of Riemanninan optimization of KL divergence, global proximities are aimed to be preserved through global distances in Poincaré disk.

a tree-like topology on the data using "reversed graph embedding" in a low-dimensional Euclidean space. However, similar to UMAP, such a representation might not exist for complex trees. SIMLR[10] is multi-kernel learning designed to perform well on datasets with multiple clusters, making it a poor choice to model data with continuous trajectories. Finally, SAUCIE[10] is an autoencoder model, which is optimized through reconstruction error, therefore its properties for preserving local and global similarities are theoretically less understood.

Recently, Ding and Regev[23] proposed an interesting follow-up on our work with a focus on eliminating the batch-correction and addressing visual crowding issues of conventional generative modeling approaches via hyperbolic embeddings. This further illustrates the benefits of hyperbolic geometry for analyzing single-cell data as proposed in this work.

## Results

Our method, Poincaré maps, is inspired by ideas from manifold learning and pseudotemporal ordering[24,25]. Given feature representations of cells such as their gene expressions, we aim to estimate the structure of the underlying tree-like manifold in three main steps (Fig. 1 and Methods): First, we compute a connected $k$-nearest-neighbor graph ($k$NNG)[26] where each node corresponds to an individual cell and each edge has a weight proportional to the Euclidean distance between the features of the two connected cells. In general, $k$NNG for a given $k$ is not necessarily connected. We need to enforce connectivity to reconstruct the hierarchy. If one component were disconnected from other components, it would be impossible to reconstruct its position relative to other components. To enforce connectivity we propose a simple procedure, described in Methods. The purpose of this first step is to estimate the local geometries of the underlying manifold, around which Euclidean distances remain a good approximation. Second, we compute global geodesic distances from the $k$NN graph, by traveling between all pairs of points along the weighted edges. This step can be computed efficiently using all pairs of shortest paths, or related measures such as the Relative Forest Accessibilities (RFA) index[27]. The purpose of this second step is to estimate the intrinsic geometry of the underlying manifold. These two first steps are commonly used in manifold learning to approximate the structure of an unknown manifold from similarities in the feature space[16,26,28,29]. As a third step, we compute a two-dimensional embedding per cell in the Poincaré disk, such that their hyperbolic distances reflect the inferred geodesic distances. The geometry of the Poincaré disk allows us to model continuous hierarchies efficiently. More specifically, embeddings that are close to the origin of the disk have a relatively small distance to all other points, representing the root of the hierarchy, or the beginning of a developmental process. On the other hand, embeddings that are close to the boundary of the disk, have a relatively large distance to all other points and are well-suited to represent leaf nodes. Thus, in Poincaré embeddings, we expect that nodes with small distances to many other nodes will be placed close to the origin of the disk. While such cells are likely from an early developmental stage, they do not necessarily belong to the root of the hierarchy (Supplementary Figs. 3–5). When a cell belonging to the root is known, we perform a translation on the Poincaré disk to place this cell in the center of the disk, easing the visualization of the hierarchy (see "Methods").

Poincaré maps have several hyperparameters to tune, such as the number of nearest neighbors ($k$), the bandwidth of the local kernel to convert distances into similarities ($\sigma$), and the scaling parameter to compute similarities in the Poincaré disk ($\gamma$). Supplementary Figs. 1–2 demonstrates the performance of Poincaré maps with respect to the choice of these hyperparameters and across different random seeds.

**Poincaré maps for single-cell analysis**. In the following, we compare Poincaré maps to prior state-of-the-art methods on various single-cell analysis tasks: visualization and lineage detection (PCA, Monocle 2[15], PAGA[8], diffusion maps[7], t-SNE[17], UMAP[18], ForceAtlas2[21], SAUCIE[12], PHATE[22] and SIMLR[10]), clustering (Louvain[30], agglomerative, k-means) and pseudotime inference (diffusion pseudotime[16]).

For this purpose, we employ multiple synthetic datasets generated from known dynamical systems and four single-cell RNA sequencing datasets varying in size, complexity (number of cell types and branches), and single-cell technology to acquire the data[2,3,31,32]. We compare Poincaré maps with the canonical hematopoietic cell lineage tree[33], and various state-of-the-art embeddings (Supplementary Note 2).

First, we evaluate the capabilities of Poincaré maps for data visualization and dimensionality reduction. It is not possible for humans to comprehend visualizations in more than three dimensions, and a third dimension already adds additional challenges for interpretation. However, for existing methods, even three-dimensional embeddings are not sufficient to capture the underlying manifold structure on many complex hierarchies. Here, we demonstrate that as few as two dimensions of Poincaré maps are enough to reconstruct the hierarchy and preserve global similarities on the datasets with a very high complexity. A common way to evaluate the quality of an embedding in labeled datasets is to use classification scores. However, this evaluation approach has limitations in the context of single-cell data, and specifically for recovering hierarchies and continuous developmental trajectories. First, quite often labels are assigned using some unsupervised learning approach, such as clustering. This could promote the dimensionality reduction method that better agrees with the label assignment method, rather than with the objective ground-truth. Second, discrete labels do not easily apply to datasets with continuous trajectories, where a clear-cut cutoff between cell types does not exist. Third, it contains no information about the quality of preservation of global similarities, e.g. positions of clusters relative to each other in the hierarchy. Instead, we use a scale-independent quality criteria[34] (see Methods and Supplementary Note 2), which was demonstrated to be a good metric to compare embeddings in an unbiased way. The criteria consist of estimating two scalar values $Q_{local}$ and $Q_{global}$ reflecting local and global properties of the dataset. We follow the assumption of Lee and Verleysen[34] that a single-cell dataset comprises a smooth manifold and a good dimensionality reduction method would preserve local and global distances on this manifold.

An important result from our experiments is that Poincaré maps is the only method that demonstrated the ability to visualize the correct branching structure of developmental processes for all datasets in terms of this quality metric (Fig. 2, Supplementary Fig. 1). Separate visual comparison of various embeddings (Supplementary Figs. 3–10) demonstrates the superior readability advantages of Poincaré maps. For example, on the dataset Paul et al.[2] only Poincaré maps and t-SNE identify the lymphoid cluster, while this important population remained invisible during exploratory data analysis when using UMAP or ForceAtlas2. Although t-SNE visualizes separate clusters well for Paul et al.[2] dataset, it disregards the hierarchical structure between clusters (see also the example in Supplementary Fig. 7). Knowledge of the position of a newly identified cluster in the developmental hierarchy could be further exploited for assigning labels (e.g. "lymphoid population") or, when the population was not known, for designing experiments to test morphological properties. Finally, Poincaré maps place the 16Neu cluster downstream of 15Mo in the hierarchy—in contrast to the canonical hierarchy, where neutrophils and monocytes are

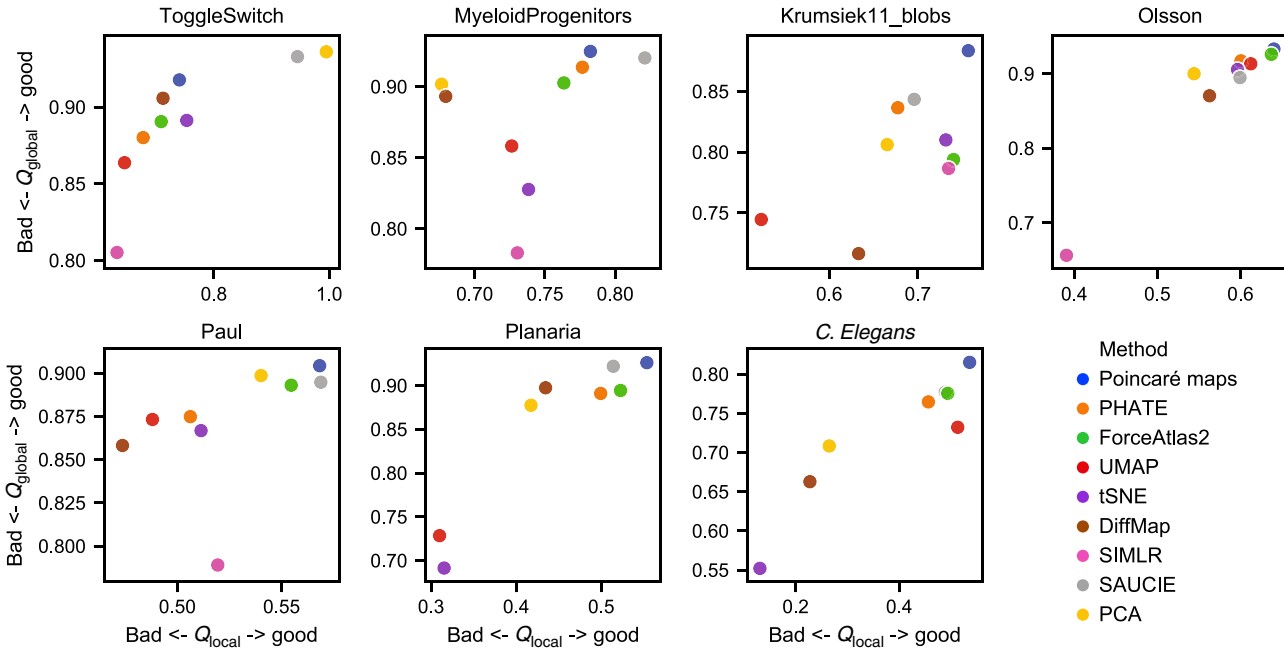

**Fig. 2 Comparison of embedding quality metric (best case) for various datasets.** Poincaré maps perform consistently well on all synthetic and real-world datasets in our evaluation. Planaria and *C. elegans* datasets—which exhibit the highest complexity in terms of number of branching trajectories—are datasets where Poincaré maps perform significantly better than other methods.

located at the same level. This result is in line with the analysis of Wolf et al.[8], indicating that the inconsistency is due to a faulty labeling of the clusters.

A back-to-back comparison of a quality metric with visual inspection gives a good intuition about the meaning of the metric quality scores and embedding properties. In particular, on datasets with simple trajectories (e.g., ToggleSwitch, MyeloidProgenitors), methods such as PCA or SAUCIE show strong performance since they preserve local similarities well. However, the performance of these methods drops significantly for the datasets with many different cell types and branches, such as Planaria and *C. elegans*. Poincaré maps in their turn, do not suffer from this limitation (Fig. 2) and significantly outperforms other methods in terms of both local and global metrics. This allows us to summarize the whole *C. elegans* cell atlas in a single Poincaré maps embedding (Fig. 3). This was not possible with UMAP with any choice of parameters, as reported by the authors in their original study. This makes Poincaré maps a strong candidate for visualization of single-cell atlases. Additional analysis of the age of the embryo on Poincaré maps revealed two distinct populations of germlines. One of these subpopulations is placed close to the border of the disc and closer to mature cell types, which potentially reflects transcription diversity of this subpopulation from other cells at the early stages. The second subpopulation is close to other cells at the early stage. We randomly picked up a cell from the second subpopulation and assigned it as a root. Fig. 3c demonstrates the relative positioning of the cell types in the hierarchy and comparison of the Poincaré pseudotime to the age of the embryo. We can see that it agrees with the age of embryo quite well, except for very early stages (<130). However, lineages are not perfectly synchronized, therefore we see significant variability on the plot.

In addition to these results, we demonstrate in Supplementary Tables 1–2 that Poincaré maps could be directly applied to achieve state-of-the-art results on clustering and pseudotime inference. Notably, for pseudotime inference the results are comparable with diffusion pseudotime, but in Poincaré maps these clusters are directly accessible as distances from the root node. Therefore they are not only fast to compute given the embedding, but also allow us to intuitively interpret a Poincaré maps plot with the root node in the center of the Poincaré disk.

**Analysis of mice hematopoiesis with Poincaré maps.** As a deeper case study, we analyze the dataset of early blood development in mice, previously studied by Moignard et al.[1], using Poincaré maps. This dataset contains measurements of cells captured in vivo with qRT-PCR at different development stages: primitive streak (PS), neural plate (NP), head fold (HF), four somite GFP (Runx1) negative (4SG-) and four somite GFP positive (4SG+) (Fig. 4a). The stages correspond to different physical times of the experiment between embryonic day 7 and day 8.25. We compare our results obtained with Poincaré maps to Moignard's diffusion maps study[1], and to Haghverdi's reconstruction of diffusion pseudotime[16]. Poincaré maps provide a qualitatively different visualization of the developmental process, where we are able to visualize the whole spectrum of the heterogeneity arising from the onset of the process. Neither PCA nor diffusion maps are able to provide a visualization of this process. While Moignard's and Haghverdi's analyses suspected an asynchrony in the developmental process, neither their application of PCA or diffusion maps were able to reveal this. In particular, previous studies suggest that the split into endothelial and erythroid sub-populations happens in the head fold. Our analysis using Poincaré maps indicates that the subpopulation fate of the cells is already predefined at primitive strike. Additionally, Poincaré maps reveal a separate cluster consisting of a mixture of cells at different developmental stages (Supplementary Fig. 11). This cluster is referred to as "mesodermal" cells by Moignard et al.[1], while by Haghverdi et al.[16] considers it as the root of the developmental process. However, as we demonstrate in Supplementary Figs. 12–13, assigning this cluster as the root of the

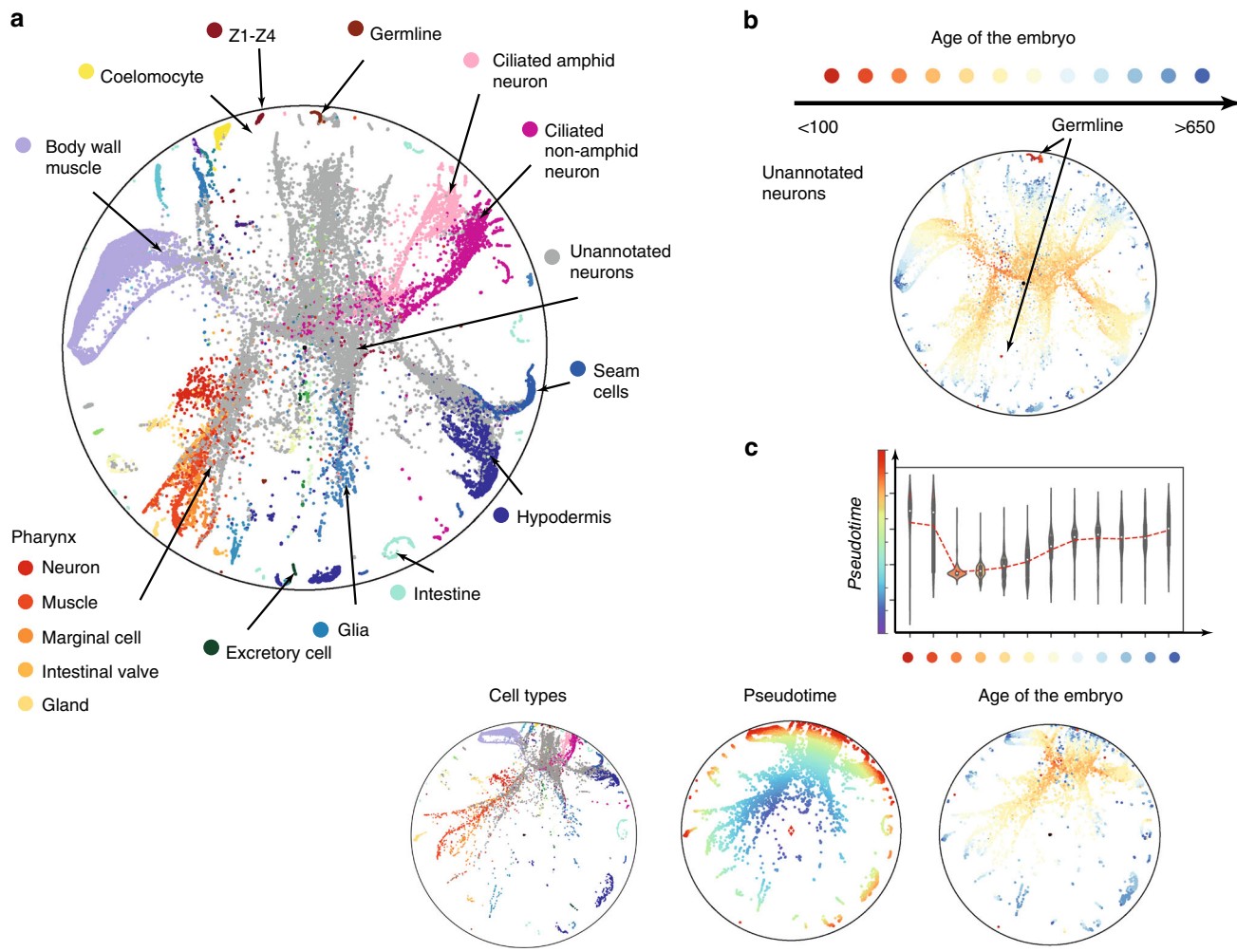

**Fig. 3 Analysis of *C. elegans* cell atlas. a** Poincaré map (without rotation) on a 40,000 cell random subsample and 100 PCA components. The parameters used for embedding are ($k = 15$, $\sigma = 2.0$, $\gamma = 3.0$). Main cell types are annotated with a text legend, the rest are separated by color. **b** Poincaré maps places mature cell types towards the border of the disk. Two subpopulations of germline cells are apparent from the embedding. **c** Rotation and comparison of Poincaré maps with respect to randomly picked up root cell form one of the sub-populations of the germlines (the one that is more similar to the rest of cell types of the early age the embryo). Red line is an average pseudotime distance for a given age of the embryo.

hierarchy would lead to a contradiction with the physical direction of time. By virtue of the Poincaré visualization, we reassigned the root of the developmental process to the furthest PS cell not belonging to the "mesodermal" cluster. We picked up a root cell from PS as to ease clustering by angle for lineage detection. More specifically, we chose the most "exterior" cell from the PS cluster, by visual inspection. Given our reassigned root, we separate the dataset into five potential lineages (see "Methods"), to find the asynchrony in the developmental process in terms of marker expressions (Fig. 4b). Analysis of the composition of cells belonging to each lineage (Fig. 4c) indicates that erythroid cells belong only to lineage 0 and this lineage contains no endothelial cells. Fig. 4d shows a substantially improved agreement of Poincaré pseudotime (with the newly reassigned root) with the experimental time (stages) compared to the pseudotime ordering proposed by Haghverdi et al.[16]. The analysis of gene expressions of main endothelial and hemogenic markers agrees with the known pattern of gene activation for endothelial and erythroid branches (Supplementary Fig. 14). Fig. 4e also demonstrates that the main hemogenic genes for the erythroid population are already expressed at the PS stage (details in Supplementary Note 3) and that the differences in gene expression apparent at all the stages between the lineages. Our analysis using Poincaré maps

suggests therefore that the fate of erythroid and endothelial cells could already be defined at primitive streak.

## Discussion

The rapid onset of popularity and accessibility of single-cell RNA sequencing technologies fueled the development of new computational approaches to analyze these data. While many computational methods exist, their results often disagree between each other. The choice of the right computational tool, done at a early stage of exploratory data analysis, will dictate the generated hypotheses about the underlying biology. Here we demonstrated that Poincaré maps reveal complex cell developmental processes that would remain undiscovered by prior methods. Poincaré maps is able to do so by leveraging hyperbolic geometry and placing minimal assumptions about the data. While any hypothesis generated via computational analysis should be validated in the lab before being converted into strong statements, a properly chosen computational tool will facilitate the selection of appropriate experiments.

For this purpose, Poincaré maps aids the discovery of complex hierarchies from single-cell data by embedding large-scale cell measurements in a two-dimensional Poincaré disk. The resulting

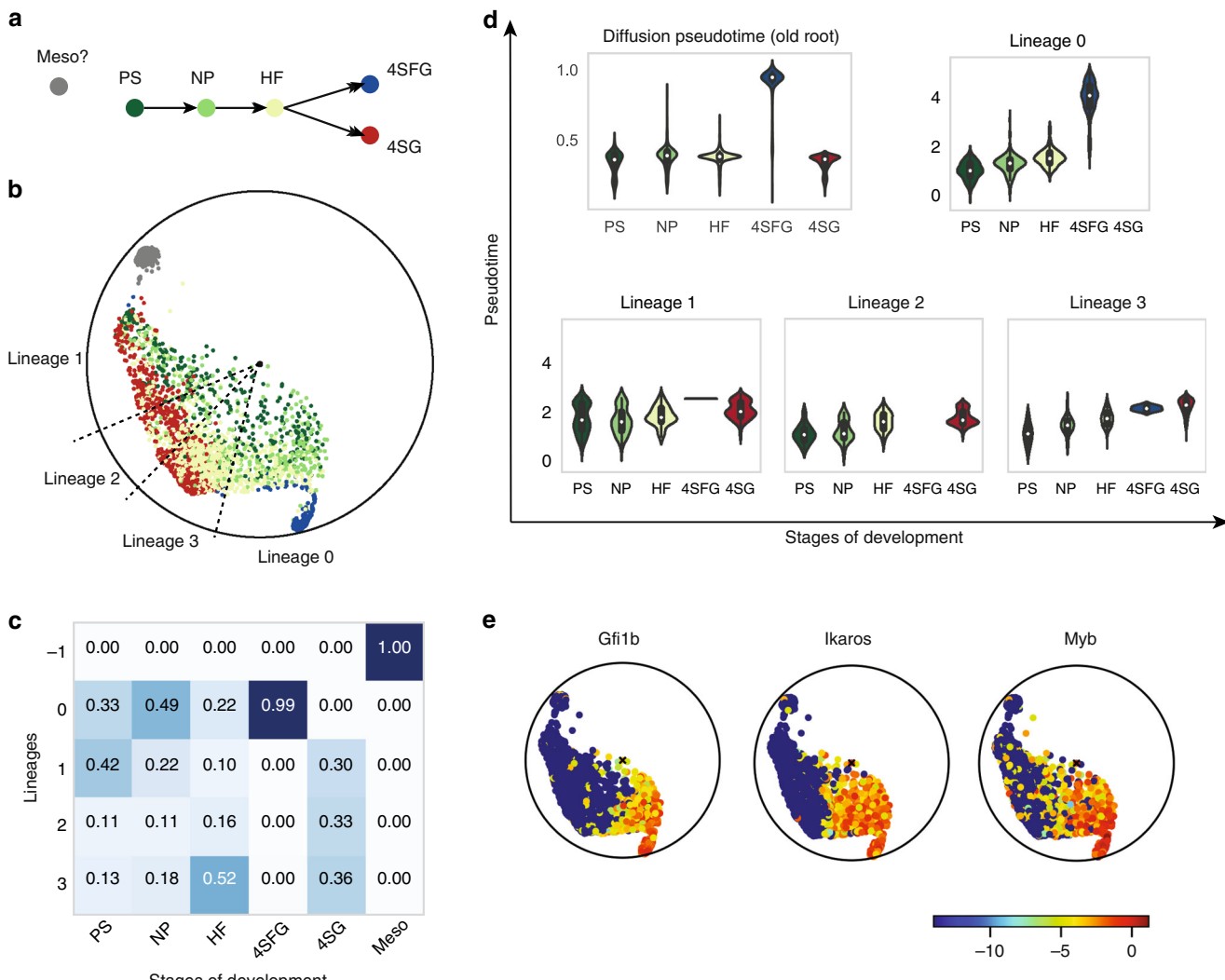

**Fig. 4 Analysis of mice hematopoiesis with Poincaré maps. a** Developmental hierarchy proposed by Moignard et al.[1] and Haghverdi et al.[16]. **b** Rotated Poincaré map with respect to reassigned root. Gray cluster represent a cluster of potential outliers or "mesodermal" cells as suggested by Moignard et al.[1]. Lineage slices were obtained with Poincaré maps (see Methods). **c** Composition of detected lineages in terms of the presence of cells from different developmental stages. **d** Violin plots of diffusion and Poincaré pseudotime for each stage of embryo developmentt. The ordering of cells proposed by Poincaré maps here has a much better agreement with developmental stages than ordering originally proposed by Haghverdi et al.[16]: we see a very clear correlation of Poincaré pseudotime with actual developmental time. **e** Gene expression of main hemogenic genes. Hemogenic genes of erythroid lineage are already expressed at the PS and NP stages.

embeddings are easy to interpret during exploratory analysis and provide a faithful representation of similarities in hierarchies due to the underlying geometry. This property makes Poincaré maps stand out among other embeddings as it allows us to simultaneously handle visualization, clustering, lineage detection, and pseudotime inference. Poincaré maps do not need to be constrained to two dimensions, and would have the same implementation for three dimensions. However, for the datasets used in this study two dimensions were sufficient; using more dimensions would reduce readability and harden interpretation.

Since Poincaré maps involves several hyperparameters and non-convex optimization, we thoroughly studied sensitivity the method performance to these parameters. Similar to most manifold learning methods, the number of nearest neighbors $k$ will have significant effect on the performance of the method. The tuning of additional hyperparameters such as $\sigma$ and $\gamma$ will have some small effect on the method's performance in terms of local and global structure, and are typically easy to select using visual inspection or the scale-independent quality measure. Finally, we

observed that the choice of random seed had no significant effect on the visualization properties.

Application of Poincaré maps is not limited to single-cell RNA sequencing. The method could be applied to any dataset on which it is possible to define a similarity measure, e.g. to any dataset to which we could apply tSNE. One example is flow and mass cytometry data[35] (CyTOF). In this paper, we focused on scRNAseq data, and leave the application of Poincaré maps to other types of data for future work. For an interested reader, we recommend consultin related methods applied to CyTOF data[36–39] to select the best preprocessing steps and local distance metric. With Poincaré maps, we hope to bring interest about hyperbolic embeddings to the biology community. Due to their advantageous properties for modeling hierarchical data, they could provide substantial benefits for a wide variety of problems such as studying transcriptional heterogeneity and lineage development in cancer from single-cell RNA and DNA sequencing data, reconstructing the developmental hierarchy of blood development, and reconstructing embryogenesis branching trajectories. We also would like to stress that

Poincaré maps could be a good candidate embedding to visualize cell atlases of whole organisms as they are able to preserve global similarities between measurements. Finally, we note that Poincaré maps are not limited to the analysis of scRNAseq, but could be applied to any type of data with a hidden hierarchical structure.

## Methods

**Data preprocessing**. First, for raw sequencing data, we strongly recommend to preprocess it with one of the preprocessing pipelines[40–43]. The pipeline should be chosen according to the sequencing machine used to produce the data[44,45]. Typically, preprocessing steps involve quality control, normalization, and log-scaling. For some datasets, batch-correction could be necessary.

Poincaré maps, like any other nearest-neighbors manifold learning method (e.g. UMAP or tSNE), can suffer from the curse of dimensionality. Therefore, when the dimensionality of data exceeds 100 dimensions, preprocessing to its 50–100 principal components is a widely used strategy to address this problem. Alternatively, methods such as scVI[14] were demonstrated to offer effective preprocessing for scRNAseq datasets, when used together with methods such as UMAP. Likewise, scVI components can be used as a preprocessing step to Poincaré maps.

For datasets with less than 100 features, PCA preprocessing is not necessary, but mean-variance normalization of individual features could be applied instead.

**Local connectivity**. Let $\mathcal{X} = \{x_i\}_{i=1}^n$ be a high-dimensional dataset of $n$ samples $\boldsymbol{x}_i \in \mathbb{R}^p$ (e.g., individual cells) with $p$ features (e.g., gene expression measurements). We first estimate local connectivity structures as typically done in manifold learning[26,28,29]. In particular, let $\mathcal{N}(i,k)$ denote the $k$ nearest neighbors of $\boldsymbol{x}_i$ in $\mathcal{X} \setminus \boldsymbol{x}_i$ according to the Euclidean distance. We then create a symmetric $k$-nearest-neighbor graph ($k$NNG) $G = (V, E, w)$, where the set of vertices $V = \{v_i\}_{i=1}^n$ represents the samples in $\mathcal{X}$ and the set of edges $E = \{v_i \sim v_j : i \in \mathcal{N}(j,k) \wedge j \in \mathcal{N}(i,k)\}$ represent the nearest-neighbor relations. In order to construct a connected $k$NNG we adopt a greedy procedure. First, we build a standard $k$NNG for a given $k$. Then, for each pair of disconnected components (if any) we find the edge with the minimum length that would connect these two components. Then, we connect the two components that can be linked using the smallest edge. We repeat this process until the kNNG has only one connected component. Furthermore, each nearest-neighbor relation is weighted using the Gaussian kernel over distances

$$w(i,j) = \begin{cases} \exp\left(-\frac{\|\boldsymbol{x}_i - \boldsymbol{x}_j\|_2^2}{2\sigma^2}\right) & \text{if } i \sim j \in E, \\ 0 & \text{otherwise,} \end{cases} \quad (1)$$

where $\sigma$ is a hyperparameter that controls the kernel width. By enforcing connectivity of $G$, we preserve finite distances between all measurements.

**Global proximities**. To estimate the underlying manifold structure from distances on the $k$NN graph $G$, we can employ all pairs shortest paths or related methods such as the Relative Forest Accessibility (RFA) index, which is defined as follows: Let $L = D - A$ denote the graph Laplacian of the graph $G$, where $A_{ij} = w(i,j)$ is the corresponding adjacency matrix and $D_{ii} = \sum_j w(i,j)$ is the degree matrix. The RFA matrix $P$ is then given as[27]

$$P = (I + L)^{-1}. \quad (2)$$

$P$ is a doubly stochastic matrix where each entry $p_{ij}$ corresponds to the probability that a spanning forest of $G$ includes a tree rooted at $i$ which also includes $j$ (i.e., where $j$ is accessible from $i$)[27,46] Compared to shortest paths, the RFA index has the advantage to increase the similarity between nodes that belong to many shortest paths. This can provide an important signal to discover hierarchical structures as nodes that participate in many shortest paths are likely close to the root of the hierarchy. In all experiments, we use the RFA index to estimate global proximities.

**Hyperbolic embedding**. Given $P$, we aim at finding an embedding $\boldsymbol{y}_i$ of each $\boldsymbol{x}_i$ that highlights the hierarchical relationships between the samples. For this purpose, we embed $P$ into two-dimensional hyperbolic space.

The Poincaré disk is the Riemannian manifold $\mathcal{P} = (\mathcal{B}, d_p)$, where $\mathcal{B} = \{\boldsymbol{y} \in \mathbb{R}^2 : \|\boldsymbol{y}\| < 1\}$ is the open 2-dimensional unit ball. The distance function on $\mathcal{P}$ is then defined as

$$d_p(\boldsymbol{y}_i, \boldsymbol{y}_j) = \text{arcosh}\left(1 + 2\frac{\|\boldsymbol{y}_i - \boldsymbol{y}_j\|^2}{(1 - \|\boldsymbol{y}_i\|^2)(1 - \|\boldsymbol{y}_j\|^2)}\right). \quad (3)$$

It can be seen from Eq. (3), that the Euclidean distance within $\mathcal{B}$ is amplified smoothly with respect to the norm of $\boldsymbol{y}_i$ and $\boldsymbol{y}_j$. This property of the distance is key for learning continuous embeddings of hierarchies. For instance, by placing the root node of a tree at the origin of $\mathcal{B}$, it would have a relatively small distance to all other nodes, as its norm is zero. On the other hand, leaf nodes can be placed close to the boundary of the disk, as the distance between points grows quickly with a norm close to one.

To compute the embedding we use an approach similar to t-SNE[17] and approximate the RFA probabilities in $P$ via distances in the embedding space. In particular, we define the similarity $q_{ij}$ between the embeddings $v_i$ and $v_j$ as

$$q_{ij} = \frac{\exp(-d_p(\boldsymbol{y}_i, \boldsymbol{y}_j)/\gamma)}{\sum_k \exp(-d_p(\boldsymbol{y}_i, \boldsymbol{y}_k)/\gamma)}, \quad (4)$$

where $\boldsymbol{y}_i, \boldsymbol{y}_j \in \mathcal{P}$. A natural measure for the quality of the embedding is then the symmetric Kullback–Leibler divergence between both probability distributions:

$$\mathcal{L}(P; \mathcal{Y}) = \sum_i \text{KL}(P_i \| Q_i) + \text{KL}(Q_i \| P_i) \quad (5)$$

**Details on the optimization**. To compute the embeddings, we minimize Eq. (5) via Riemannian Stochastic Gradient Descent (RSGD)[47]. In particular, we update the embedding of $y_i$ in epoch $t$ using

$$\boldsymbol{y}_i^{t+1} \leftarrow \mathfrak{R}_{\boldsymbol{y}_i^t}(-\eta \text{grad}(\mathcal{L}, \boldsymbol{y}_i^t)), \quad (6)$$

where $\text{grad}(\mathcal{L}, \boldsymbol{y}_i^t)$ denotes the Riemannian gradient of Eq. (5) with respect to $\boldsymbol{y}_i^t$, $\mathfrak{R}_{\boldsymbol{y}_i^t}$ denotes a retraction (or the exponential map) from the tangent space of $\boldsymbol{y}_i^t$ onto $\mathcal{P}$, and $\eta > 0$ denotes the learning rate. The optimization can be performed directly in the Poincaré disk or, alternatively, in the Lorentz model of hyperbolic space which provides improved the numerical properties and efficient computation of the exponential map[48].

**Translation in $\mathcal{P}$**. Eq. (5) favors embeddings where nodes with short distances to all other nodes are placed close to the origin of the disk. While such nodes correspond often to nodes that are close to the root of the underlying tree, it is not guaranteed that the root is the closest embedding to the origin. However, when the root node is known, we can perform an isometric transformation of the entire embedding that places this node at the origin and preserves all distances between the points. In particular, to translate the disk such that the origin of the Poincaré disk is translated to $\boldsymbol{v}$, $\boldsymbol{x}$ is translated to

$$\tau(\boldsymbol{x}, \boldsymbol{v}) = \frac{(1 + 2\langle \boldsymbol{v}, \boldsymbol{x} \rangle + \|\boldsymbol{x}\|^2)\boldsymbol{v} + (1 - \|\boldsymbol{v}\|^2)\boldsymbol{x}}{1 + 2\langle \boldsymbol{v}, \boldsymbol{x} \rangle + \|\boldsymbol{v}\|^2 \|\boldsymbol{x}\|^2} \quad (7)$$

Since the spatial resolution is amplified close to the origin of the disk, provides also a method to zoom into different parts of the embedding by moving the area of interest to the origin.

**Clustering**. Hyperbolic space is a metric space and thus allows us to compute distances between any pair of points. This makes Poincaré maps straightforwardly applicable to clustering techniques that rely only on pairwise (dis)similarity measurements such as spectral clustering, agglomerative clustering, and kmedoids.

**Lineages**. As a naive approach for lineage detection, we suggest using agglomerative clustering by the angle between a pair of points in the Poincaré disk after the rotation with respect to the root node.

**Poincaré pseudotime**. "Pseudotime" is typically referred as "a measure of how much progress an individual cell has made through a process such as cell differentiation"[25]. As Poincaré pseudotime we propose to use the distance from the root node in the Poincaré disk.

**Choice of hyperparameters**. In the following, we discuss the function of different hyperparameters in Poincaré maps and propose typical value ranges. The number of nearest neighbors $k$ reflects the average connectivity of the clusters and is typically set to $k = 15, 20, 30$. The Gaussian kernel width $\sigma$ is responsible for the weights for the $k$-NN graph in the original space and is typically set to $\sigma = 1.0, 2.0$. The softmax temperature $\gamma$ controls the scattering of embeddings and is typically set to $\gamma = 1.0, 2.0$.

**Scale-independent quality measure**. To quantitatively compare the performance of different embedding approaches, we use the scale-independent quality criteria proposed by Lee and Verleysen[34] The main idea of this approach is that a good dimensionality reduction approach, will have good preservation of local and global distances on the manifold, e.g. close neighbors should be placed close to each other while maintaining large distances between distant points. Lee and Verleysen[34] proposed to use two scalar quality criteria $Q_{\text{local}}$ and $Q_{\text{global}}$ focusing separately on low and high-dimensional qualities of the embedding. The quantities of $Q_{\text{local}}$ and $Q_{\text{global}}$ range from 0 (bad) to 1 (good) and reflect how well are local and global properties of the dataset are preserved in the embedding (see details in Supplementary Note 2). To estimate distances in the high-dimensional space $\delta_{ij}$, we use geodesic distances estimated as the length of a shortest-path in a $k$-nearest neighbors graph. We fixed $k = 20$ for all the datasets as there is no objective way to decide on a correct $k$ and visual results looked good for all the embeddings for this choice of $k$. For the distances low-

dimensional space we use euclidean space for all the embeddings except Poincaré maps, for which we use hyperbolic distances. As all the embeddings involve an element of stochasticity in their output, we run every embedding three times with a different seed. We run all the embeddings with a different set of parameters.

**Computational complexity and time.** Memory complexity of Poincaré maps is $O(n^2)$, where $n$ is the number of samples. Time complexity consist of three parts: estimation of $k$NNG – $O(n^2)$ (this part could be replaced with FAISS[49] for scalability), estimation of RFA – $O(n^2)$ and minimization of KL divergence – $O(neb)$, where $e$—maximum number of epochs, $b$—batch size. As we need to minimize KL till convergence, we can in advance estimate the number of epochs needed. For all the datasets used here, the number of epochs was less than 2000 and we also used early stopping upon convergence. Typical running time on 1 GPU for all the small-medium datasets is less than a minute, and for large datasets around 15 min (Planaria) or 2–3 h (C. elegans).

**Reporting summary.** Further information on research design is available in the Nature Research Reporting Summary linked to this article.

## Data availability
Several public datasets were used in this study: three synthetic datasets generated with Scanpy, Olsson et al.[3] (synapse ID https://www.synapse.org/#Synapse: syn4975060syn4975060), Paul et al.[2] (accession code https://www.ncbi.nlm.nih.gov/geo/query/acc.cgi?acc=GSE72857GSE72857), Moignard et al.[1] (accession code https://www.ncbi.nlm.nih.gov/geo/query/acc.cgi?acc=GSE61470GSE61470), Plass et al.[31] (accession code https://www.ncbi.nlm.nih.gov/geo/query/acc.cgi?acc=GSE103633GSE103633, preprocessed data available at https://shiny.mdc-berlin.de/psca/), Packer et al.[32] (preprocessed data available at https://github.com/qinzhu/VisCello).

## Code availability
The code is available at https://github.com/facebookresearch/PoincareMaps.

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

## Acknowledgements

The authors would like to thank Ioana Sandu and Will Macnair for valuable discussions, Nafissa Yakubova for the support of the project and Tim Lierman for the help in the design of Fig. 1.

## Author contributions

L.B., M.N., and A.K. conceived the idea. A.K., M.N., and D. L.-P. designed and implemented the computational tools. A.K. performed the analysis and biological interpretation of results. L.B. contributed to the design of the study. M.N. supervised the study. A.K., M.N., and D.L.-P. wrote the manuscript.

## Competing interests

The authors declare no competing interests.
