## [Peer Review File · Nature Communications]

Reviewers' comments:

Reviewer #1 (Remarks to the Author):

In this work, Klimovskaia et al. introduce Poincaré maps for the analysis of single-cell data. The overall idea is to compute geodesic cell-cell distances over a k-NN graph and optimize the resulting embedding so that the points on the Poincaré disk respect the cell-cell distances. They go on to demonstrate their method on a variety of single-cell datasets on tasks such as visualization, pseudotime ordering, and clustering and compare to major related methods.

Overall, the concept is novel in the context of single-cell data analysis and is a promising approach for visualizing branching structures of single-cell data. However, as it currently stands the manuscript is critically underdeveloped with respect to (i) comparisons to alternative and simple baseline methods, (ii) quantitative justification for claims made, and (iii) details to reproduce the work as presented. Details below.

Major

The authors claim "Poincaré maps was the only method that demonstrated the ability to visualize the correct branching structure..." compared to algorithms like tSNE and UMAP. Firstly, I think this claim is debatable (e.g. SFig 5 doesn't pick up myelocytes follow gran cells). Secondly, the major concern is that tSNE and UMAP are treated as single results, but in fact are highly sensitive to random seed initialization and hyperparameter settings. However, the question of random seed is not addressed. Furthermore, to select parameters the authors write "If no parameters were provided, we performed a parameter search to achieve the best performance for each method." -- however, I cannot work out how a rigorous parameter search is performed when the outcome is subjective (how well the visualizations capture the lineage). The authors should attempt to construct a quantitative measure of this in order to conduct a fair comparison, and note that the default parameters for e.g. a tSNE implementation may not lead to a good result for single-cell data.

Similarly, Poincaré maps employs stochastic gradient descent for a non convex optimization problem, so is presumably sensitive to initialization and random seed. The authors should demonstrate the robustness of the embeddings with respect to these. There are several free parameters of method (σ and γ), but the authors only say they are "typically" set to a certain range, or for the number of nearest neighbours (k) no details are provided. They should (i) provide guidance on how to set these parameters, and (ii) demonstrate robustness of the embeddings to variation in these.

In the section "Poincaré maps robustly predict values on unseen intermediate types", the evaluation is quantitatively lacking. The authors state that they "perform this task 1.3-3 times better than other embedding methods" but it is not clear with respect to what metric or methods. This should be reported with respect to e.g. mean square error of predicted vs ground truth, with details spelled out for all methods/datasets, etc. Similarly, Figure 4b-c is not well explained, e.g. why are PC values being compared rather than expression values? What is the colour scale on figure 4B?

In the section "Poincaré maps robustly predict values on unseen intermediate types" several simple baseline comparisons are missing. In particular, if x is the (vector of) expression values of the root cell and y is the (vector of) of the cells at the leafs (differentiated), we can linearly interpolate and "predict" intermediate stages via linear interpolation of $px + (1-p)y$ for $p \in [0,1]$. Similarly, the authors should include PCA in the current setup rather than comparing to simply ForceAtlas2 and UMAP, for a variety of PCs.

As far as I can tell the authors compare to tSNE, UMAP, and ForceAtlas2, but there is a large

literature of single-cell RNA-seq visualization methods and I feel it is unfair to leave the majority of these out of comparisons. Examples include Phate [1], SIMLR [2], scVI [3], SAUCIE [4], scVis [5], etc. The authors should attempt to at least compare to some of these.

[1] <https://www.biorxiv.org/content/10.1101/120378v4>

[2] <https://www.nature.com/articles/nmeth.4207>

[3] <https://www.nature.com/articles/s41592-018-0229-2>

[4] <https://openreview.net/pdf?id=ryV9NFyvM>

[5] <https://www.nature.com/articles/s41467-018-04368-5>

The section "Poincaré maps generate new hypothesis about early blood development in mice" essentially makes a hypothesis about when endothelial and erythroid sub-populations emerge but without external validation this is just a hypothesis that does not support the validity of the method. The authors claim that the "main hemogenic genes for the erythroid population are already expressed at the PS stage" as evidence, but surely this can be inferred from simply plotting the expression of the hemogenic genes as a function of stage without any of the Poincaré analysis?

Relatedly, the authors state that "by virtue of the Poincaré visualization, we reassigned the root of the developmental process to the furthest PS cell not belonging to the 'mesodermal' cluster". However, from looking at supplementary figure 13 it is highly non-obvious how they came to this conclusion, nor that it is necessarily the correct thing to do since presumably a mesodermal cell type would be at the root of the differentiation trajectory? The authors should expand on their reasoning behind this.

Minor

Definition of Poincare disk be in R^n rather than R^2 or be 2 dimensional unit ball

Just before equation 4 redefines the embeddings as v rather than y

"Aparent" misspelled page 6

Reviewer #2 (Remarks to the Author):

Anna Klimovskaia et al. proposed a new model, called Poincaré maps, for the visualization, clustering, pseudotime inference, and unseen cell prediction for single cell RNA-seq datasets. The main novelty of their methods is the identification of hierarchical structures, which could be important for the study of developmental biology. The authors compare their methods with some state-of-art algorithms, such as t-SNE, UMAP, PAGA, and Monocle, on some synthetic and real datasets. The method is largely based on their original Poincaré embedding algorithm proposed mainly for word embedding (Maximilian Nickel, et al., 2017). However, I don't think the article is currently outstanding enough to publish in Nature Communications. Below are my major concerns.

1. I noticed the hyperbolic embedding method has already been published in the author's another paper. To adapt this embedding method to scRNA-seq datasets,

(a) Are only kNN graph construction and global proximities to measure the distance between cells key steps for the adaptation? If so, it will be better to discuss more about the importance of these steps.

(b) Some guidelines for the application on different datasets could be discussed, such as different scRNA-seq protocols (such as the 10x Genomics), and as the authors claimed in the Introduction section, the CYTOF data. For example, how should the data be transferred to the embedding algorithm?

(c) And how about the time cost with respect to datasets with large number of cells?

2. Construction of kNN graph. I think this is a critical step for the success of the proposed method as in some other scRNA-seq algorithms. However, the authors fail to make their operations clear enough.

(a) How about the robustness with respect to the choice of k and σ ?

(b) Why and how does the author enforce the connectivity of the kNN graph? Is it for the definition of RFA? If so, is it reasonable for datasets with obviously separable cell types?

(c) The distance used to discover the nearest neighbors is also ambiguous (the author claims the use of PCA in some datasets in the Supplementary but not in the Methods section).

(d) In addition, kNN graphs constructed are usually noisy, as described in other algorithms (such as Stuart, T., et al. Cell, 2019; Bendall, S.C., et al. Cell, 2014; Manu Setty, et al. Nat Biotechnology, 2016). Does this have effects on the performance?

3. As far as I know, there may be a lot of models of the hyperbolic space, including hyperboloid, Klein, Lorentz, etc. Can the author discuss some concerns for their selection of Poincaré disk?

4. For the unseen cell prediction task, are there some requirements for the plausible prediction? For example, what kinds of properties of the unseen cell types should have for their prediction?

5. Benchmark. I noticed all real datasets tested are published before 2018. Can the author test the method on some 'fashionable' datasets? Can the author compare the method with some recently published algorithm, for example, scVI, which is also based on the neural network, as the author's prediction strategy?

6. I suggest the 'Introduction' and 'Discussion' parts should be re-organized to clarify the significance and include some in-depth discussion.

7. Some figures should be better selected and annotated. For example, I don't think Figure 1 (a) (b) are necessary for the main text. Colors on some figures (Fig1 (b) (e)) are not easy to understand, or may be meaningless. Point sizes are too large in Fig2, making the visualization very disordered. The legends of fig 2 emphasize the identification of an outlier cluster. How about the hierarchy? Figure 3 and Figure 4 can be combined. For figure 4, the figure legends are not enough for the interpretations. What do the colored circles of the row labels of figure 4 (b) mean? What do the values of heatmap (again fig 4 b) mean and how are these values computed? I think all figure legends should be rewritten for better interpretation.

8. There are a lot of syntax errors, such as:

(a) Page 2, line 4 (main text) 'what is essential'

(b) Page 2, figure 1 legend, both (d) and (e) have some mistakes

(c) Page 2, 'Results' line 3 'where each node ... and where each edge ...'

(d) Page 3, 'PAGA produce'

(e) Page 4, line 3 'The limitation of this approach we demonstrate'

(f) Page 6, 'Discussion' 'This property make'

(g) Page 8, Five lines from the bottom, 'Since..., provides'

I believe this list is not exhaustive, and the author should totally polish their words.

We would like to thank the anonymous reviewers for their valuable and in-depth comments. We have addressed all of the comments, improving substantially the quality of the paper. Below we provide a detailed reply to the questions raised by the reviewers with the references to the corresponding sections in the main text and the supplementary. In the main text and supplementary the changes are highlighted in red.

Reviewer #1 (Remarks to the Author):

In this work, Klimovskaia et al. introduce Poincaré maps for the analysis of single-cell data. The overall idea is to compute geodesic cell-cell distances over a k-NN graph and optimize the resulting embedding so that the points on the Poincaré disk respect the cell-cell distances. They go on to demonstrate their method on a variety of single-cell datasets on tasks such as visualization, pseudotime ordering, and clustering and compare to major related methods.

Overall, the concept is novel in the context of single-cell data analysis and is a promising approach for visualizing branching structures of single-cell data.

However, as it currently stands the manuscript is critically underdeveloped with respect to

- (i) comparisons to alternative and simple baseline methods,
- (ii) quantitative justification for claims made
- (iii) details to reproduce the work as presented. Details below.

Major

The authors claim “Poincaré maps was the only method that demonstrated the ability to visualize the correct branching structure...” compared to algorithms like tSNE and UMAP.

Firstly, I think this claim is debatable (e.g. SFig 5 doesn't pick up myelocytes follow gran cells).

Actually, myelocytes were following gran in the old image, but because of the hyperbolic distances and choice of the colors for the embedding, probably it was difficult to notice. To avoid further similar confusions, we added labels to the plots. Now the position of the label is chosen as medoids of a corresponding cluster with respect to hyperbolic distances.

Secondly, the major concern is that tSNE and UMAP are treated as single results, but in fact are highly sensitive to random seed initialization and hyperparameter settings. However, the question of random seed is not addressed. Furthermore, to select parameters the authors write “If no parameters were provided, we performed a parameter search to achieve the best performance for each method.” -- however, I cannot work out how a rigorous parameter search is performed when the outcome is subjective (how well the visualizations capture the lineage).

The authors should attempt to construct a quantitative measure of this in order to conduct a fair comparison, and note that the default parameters for e.g. a tSNE implementation may not lead to a good result for single-cell data.

Similarly, Poincaré maps employs stochastic gradient descent for a non convex optimization problem, so is presumably sensitive to initialization and random seed. The authors should demonstrate the **robustness of the embeddings with respect to these**. There are several free parameters of method (σ and γ), but the authors only say they are “typically” set to a certain range, or for the number of nearest neighbours (k) no details are provided. They

should (i) provide guidance on how to set these parameters, and (ii) demonstrate robustness of the embeddings to variation in these.

We added the quantitative measure and an extensive search for the best parameters through it for all the embeddings. This issue now reflects in the main text, Figure 2 and additional details in Supplementary Note 2. We also added the study of robustness to the Methods section and additional discussion to the main text and to Supplementary Note 2.

In the section “Poincaré maps robustly predict values on unseen intermediate types”, the evaluation is quantitatively lacking. The authors state that they “perform this task 1.3-3 times better than other embedding methods” but it is not clear with respect to what metric or methods. This should be reported with respect to e.g. mean square error of predicted vs ground truth, with details spelled out for all methods/datasets, etc. Similarly, Figure 4b-c is not well explained, e.g. why are PC values being compared rather than expression values? What is the colour scale on figure 4B?

In the section “Poincaré maps robustly predict values on unseen intermediate types” several simple baseline comparisons are missing. In particular, if x is the (vector of) expression values of the root cell and y is the (vector of) of the cells at the leafs (differentiated), we can linearly interpolate and “predict” intermediate stages via linear interpolation of $px + (1-p)y$ for $p \in [0, 1]$. Similarly, the authors should include PCA in the current setup rather than comparing to simply ForceAtlas2 and UMAP, for a variety of PCs.

- Add linear baseline
- Add VAE baseline
- Add PCA baseline
- Add scVI baseline

We thank reviewer 1 for pointing out on a very important overlook on our side. While we focused on demonstrating the superiority of Poincaré maps compared to advanced methods such as tSNE, UMAP and ForceAtlas2, we forgot to benchmark it against a simple linear baseline. Indeed, after a careful inspection, it appeared that the trajectories of the removed cells were be monotonic, so a simple linear baseline demonstrates very good performance. The fact that Poincaré maps were able to reconstruct it, confirm the properties of preserving local and global distances in the low dimensional representation and its advantage compared to other dimensionality reduction methods. However, since the linear baseline (linear interpolation in high dimensions) performs equally well, we don't consider this section interesting and significant anymore. Therefore, we removed all the text and results regarding the interpolation. Answering the previous question of how we quantitatively compared the interpolation this information was in the supplementary. We used dynamic time warping distance for the reconstructed pseudo temporal ordering (diffusion pseudotime).

As far as I can tell the authors compare to tSNE, UMAP, and ForceAtlas2, but there is a large literature of single-cell RNA-seq visualization methods and I feel it is unfair to leave the majority of these out of comparisons. Examples include Phate [1], SIMLR [2], scVI [3], SAUCIE [4], scVis [5], etc. The authors should attempt to at least compare to some of these.

[1] <https://www.biorxiv.org/content/10.1101/120378v4>

[2] <https://www.nature.com/articles/nmeth.4207>

[3] <https://www.nature.com/articles/s41592-018-0229-2>

[4] <https://openreview.net/pdf?id=ryV9NFyvM>

[5] <https://www.nature.com/articles/s41467-018-04368-5>

We added comparison to SAUCIE, Phate and SIMLR. scVI uses UMAP for visualization in 2 dimensions and doesn't work with the format of data we have (typically we used PCA components). However, it could be used to replace PCA (n_PCA=50) for our preprocessing before we compute the Poincaré maps, but this is not relevant to the actual goal and advancements of Poincaré maps algorithm. We could not make the code for scVIs work.

The section "Poincaré maps generate new hypothesis about early blood development in mice" essentially makes a hypothesis about when endothelial and erythroid sub-populations emerge but without external validation this is just a hypothesis that does not support the validity of the method.

Yes, this is a hypothesis. However, in the text we provide a proof (demonstration) of why the alternative hypothesis cannot be correct.

The authors claim that the "main hemogenic genes for the erythroid population are already expressed at the PS stage" as evidence, but surely this can be inferred from simply plotting the expression of the hemogenic genes as a function of stage without any of the Poincaré analysis?

We used this fact as validation. However, this fact was not checked in both original studies on the same dataset. We suppose that a misleading branching structure produced by diffusion maps led the authors to the formulation of the wrong hypothesis. Therefore, they never checked that the erythroid population was already expressed at the PS stage.

Relatedly, the authors state that "by virtue of the Poincaré visualization, we reassigned the root of the developmental process to the furthest PS cell not belonging to the 'mesodermal' cluster". However, from looking at supplementary figure 13 it is highly non-obvious how they came to this conclusion, nor that it is necessarily the correct thing to do since presumably a mesodermal cell type would be at the root of the differentiation trajectory? The authors should expand on their reasoning behind this.

We added the details to the main text.

Minor

Definition of Poincare disk be in R^n rather than R^2 or be 2 dimensional unit ball

Changed in Methods section.

**Just before equation 4 redefines the embeddings as v rather than y
 *y defines coordinates and v denotes edges.***

**"Aparent" misspelled page 6
*Fixed.***

Reviewer #2 (Remarks to the Author):

Anna Klimovskaia et al. proposed a new model, called Poincaré maps, for the visualization, clustering, pseudotime inference, and unseen cell prediction for single cell RNA-seq datasets. The main novelty of their methods is the identification of hierarchical structures, which could be important for the study of developmental biology. The authors compare their methods with some state-of-art algorithms, such as t-SNE, UMAP, PAGA, and Monocle, on some synthetic and real datasets. The method is largely based on their original Poincaré embedding algorithm proposed mainly for word embedding (Maximilian Nickel, et al., 2017). However, I don't think the article is currently outstanding enough to publish in Nature Communications. Below are my major concerns.

1. I noticed the hyperbolic embedding method has already been published in the author's another paper. To adapt this embedding method to scRNA-seq datasets,
 - (a) Are only kNN graph construction and global proximities to measure the distance between cells key steps for the adaptation? If so, it will be better to discuss more about the importance of these steps.
 - (b) Some guidelines for the application on different datasets should be discussed, such as different scRNA-seq protocols (such as the 10x Genomics), and as the authors claimed in the Introduction section, the CYTOF data. For example, how should the data be transferred to the embedding algorithm?
 - (c) And how about the time cost with respect to datasets with large number of cells?

We rewrote the main text and Methods sections to address these questions.

2. Construction of kNN graph. I think this is a critical step for the success of the proposed method as in some other scRNA-seq algorithms. However, the authors fail to make their *operations clear enough*.

- (a) How about the robustness with respect to the choice of k and sigma?
- (b) Why and how does the author enforce the connectivity of the kNN graph? Is it for the definition of RFA? If so, is it reasonable for datasets with obviously separable cell types?
- (c) The distance used to discover the nearest neighbors is also ambiguous (the author claims the use of PCA in some datasets in the Supplementary but not in the Methods section).
- (d) In addition, kNN graphs constructed are usually noisy, as described in other algorithms (such as Stuart, T., et al. Cell, 2019; Bendall, S.C., et al. Cell, 2014; Manu Setty, et al. Nat Biotechnology, 2016). Does this have effects on the performance?*

We added a study on robustness to the main text and to the Supplementary Note 2.

3. As far as I know, there may be a lot of models of the hyperbolic space, including hyperboloid, Klein, Lorentz, etc. Can the author discuss some concerns for their selection of Poincaré disk?

This is now discussed in the introduction.

4. *For the unseen cell prediction task, are there some requirements for the plausible prediction? For example, what kinds of properties of the unseen cell types should have for their prediction?*
5. *Benchmark. I noticed all real datasets tested are published before 2018. Can the author test the method on some 'fashionable' datasets? Can the author compare the method with some recently published algorithm, for example, scVI, which is also based on the neural network, as the author's prediction strategy?*

This section has been removed.

6. I suggest the 'Introduction' and 'Discussion' parts should be re-organized to clarify the significance and include some in-depth discussion.

We have rewrote these sections to to clarify the significance and include more in-depth discussions.

7. **Some figures should be better selected and annotated.** For example, I don't think Figure 1 (a) (b) are necessary for the main text. Colors on some figures (Fig1 (b) (e)) are not easy to understand, or may be meaningless. **Point sizes are too large in Fig2, making the visualization very disordered.** The legends of fig 2 emphasize the identification of an outlier cluster. How about the hierarchy? Figure 3 and Figure 4 can be combined. For figure 4, the figure legends are not enough for the interpretations. What do the colored circles of the row labels of figure 4 (b) mean? What do the values of heatmap (again fig 4 b) mean and how are these values computed? I think all figure legends should be rewritten for better interpretation.

We have updated Figures 1-3 to accommodate the reviewer's comments.

8. There are a lot of syntax errors, such as:

- (a) Page 2, line 4 (main text) 'what is essential'
- (b) Page 2, figure 1 legend, both (d) and (e) have some mistakes
- (c) Page 2, 'Results' line 3 'where each node ... and where each edge ...'
- (d) Page 3, 'PAGA produce'
- (e) Page 4, line 3 'The limitation of this approach we demonstrate'
- (f) Page 6, 'Discussion' 'This property make'
- (g) Page 8, Five lines from the bottom, 'Since..., provides'

I believe this list is not exhaustive, and the author should totally polish their words.

** See Nature Research's author and referees' website at www.nature.com/authors for information about policies, services and author benefits

This email has been sent through the Springer Nature Tracking System NY-610A-NPG&MTS

<https://github.com/KrishnaswamyLab/PHATE>

<https://github.com/KrishnaswamyLab/SAUCIE>

<https://github.com/scvae/scvae>

<https://github.com/YosefLab/scVI>

<https://github.com/BatzoglouLabSU/SIMLR>

Reviewers' comments:

Reviewer #1 (Remarks to the Author):

The manuscript is much improved. However, the authors write

"We added the quantitative measure and an extensive search for the best parameters through it for all the embeddings. This issue now reflects in the main text, Figure 2 and additional details in Supplementary Note 2. We also added the study of robustness to the Methods section and additional discussion to the main text and to Supplementary Note 2."

which was in reference to my asking that they acknowledge that other methods (UMAP, tSNE, etc) have tunable parameters and benchmark against multiple realizations for each. However, I cannot find any details of how this "extensive search" is performed and with regards to what parameters (for the methods other than Poincaré Embeddings) -- even Ctrl+F for "search" in the main manuscript and supplement turns up nothing relevant. It is crucial the authors detail this for the manuscript to be reproducible.

Reviewer #2 (Remarks to the Author):

The authors may possibly answer some of my questions, but they only "asked" me to check their response in their corrected manuscript by myself, with limited clues for me. So I am not very sure the authors gave enough response. By re-checking their main text, I think they ignored some of my comments. For example, they just removed the description of Cytof data in the Introduction section (which was claimed in their original version). For my concern of connectivity, the authors added one sentence "To enforce connectivity we propose a simple procedure, described in Online Methods." in the main text. But I don't find any such procedure in the Methods section. The authors gave no response for the comparison with scVI.

Finally, I noticed there was a recent paper called "Deep generative model embedding of single-cell RNA-Seq profiles on hyperspheres and hyperbolic spaces" (<https://www.biorxiv.org/content/10.1101/853457v1>). They developed a method called 'scSphere', which also leveraged the Poincaré disk. Is it possible to make some comparisons?

Reviewer #1 (Remarks to the Author):

The manuscript is much improved. However, the authors write

"We added the quantitative measure and an extensive search for the best parameters through it for all the embeddings. This issue now reflects in the main text, Figure 2 and additional details in Supplementary Note 2. We also added the study of robustness to the Methods section and additional discussion to the main text and to Supplementary Note 2."

which was in reference to my asking that they acknowledge that other methods (UMAP, tSNE, etc) have tunable parameters and benchmark against multiple realizations for each. However, I cannot find any details of how this "extensive search" is performed and with regards to what parameters (for the methods other than Poincaré Embeddings) -- even Ctrl+F for "search" in the main manuscript and supplement turns up nothing relevant. It is crucial the authors detail this for the manuscript to be reproducible.

Response to Reviewer #1:

Dear reviewer #1,

Thank you for your time and comments. The "extensive search" was added to the Supplementary Note 2 in Supplementary Figure 2 (comparison between different benchmarks) and Supplementary Figure 3 (robustness of Poincaré maps to parameters and random seed). But as you point out, we didn't specify exact parameters in the text, so we do it now. Please, note, while main Figure 2 and Supplementary Figure 2 look similar, they are not the same. In main Figure 2 we present the best performance of each method, while in Supplementary Figure 2, we show all the runs (different random seeds, all possible parameters we tested). This was indeed not very clear, so we have added more details to the Supplementary Note 2 text (page 3, we highlight it red for this review iteration). Here is the addition to the supplementary text that we made:

"For our comparisons, we used the scanpy implementation of PCA, UMAP, tSNE, and diffusion maps, as these are very effective implementations adapted for single-cell datasets. The scanpy package provides recommendations for the default set of parameters (demonstrated to work well on a wide range of single-cell datasets), so we tried all the recommended parameters. In particular, for UMAP we used $\gamma = 1.0, 2.0$, $\min_dist = 0.1, 1.0, 0.5$, $spread = 0.1, 0.5, 1.0$. For tSNE, the scanpy implementation allows to vary perplexity, but since this parameter is linked to the k in k -nearest neighbors, we fix it for all the methods for a fair comparison. For ForceAtlas2, we used a PAGA initialization (with a resolution of 0.9, as recommended by the authors) as it was

demonstrated to substantially improve the performance of the ForceAtlas2 method. Diffusion maps have only two parameters `n_comp` (number of dimensions) and `k` nearest neighbours, which are fixed between all the methods for a fair comparison. For SIMLR, we provided additional advantage by using information about the number of cell types: `nc` = number of cell types computed from annotated labels, `cores.ratio` = 0. For PHATE, we used parameters recommended by the authors in their tutorial notebook: `knn_dist="euclidean", gamma=0, t=12, decay=15`. For SAUCIE, we used `steps=1000`.”

Reviewer #2 (Remarks to the Author):

The authors may possibly answer some of my questions, but they only "asked" me to check their response in their corrected manuscript by myself, with limited clues for me. So I am not very sure the authors gave enough response. By re-checking their main text, I think they ignored some of my comments. For example, they just removed the description of Cytof data in the Introduction section (which was claimed in their original version). For my concern of connectivity, the authors added one sentence "To enforce connectivity we propose a simple procedure, described in Online Methods." in the main text. But I don't find any such procedure in the Methods section. The authors gave no response for the comparison with scVI.

Finally, I noticed there was a recent paper called "Deep generative model embedding of single-cell RNA-Seq profiles on hyperspheres and hyperbolic spaces" (<https://www.biorxiv.org/content/10.1101/853457v1>). They developed a method called 'scPhere', which also leveraged the Poincaré disk. Is it possible to make some comparisons?

Response to Reviewer #2:

Dear Reviewer #2,

We apologize for the inconvenience of referring to the main text. We did it because the comments from both reviewers were similar and we thought that it is easier for everyone to summarize it in the main text. As it is reflected in red font in the paper, thanks to your and reviewer #1 comments, we added significant changes to the paper. In this response, we add comments to your current review, as well as detailed comments to your previous review in order to avoid any misunderstanding.

Review iteration 2:

- **"they just removed the description of Cytof data in the Introduction section"** - We removed it from the main section since as you pointed out we don't have any current experiments to support this claim. However, there is no problem to run Poincaré maps on Cytof data since the method is generic and could be applied to any type of data on which we can define similarities, e.g. Poincaré maps is applicable to any dataset for which tSNE or UMAP are applicable. As you know, tSNE is very widely used for Cytof data. We propose to add the following paragraph to the discussion (main text, page 7): **"Application of Poincaré maps is not limited to single-cell RNA sequencing. The method could be applied to any dataset on which it is possible to define a**

similarity measure, e.g. to any dataset to which we could apply tSNE. One example is flow and mass cytometry data \cite{bendall2011single} (CyTOF).

In this paper we focused on scRNAseq data, and leave the application of Poincaré maps to other types of data for future work. For an interested reader, we recommend consulting related methods applied to CyTOF data \cite{setty2016wishbone, marco2014bifurcation, qiu2011extracting, bendall2014single} to select the best pre-processing steps and local distance metric.”

- **“concern of connectivity”**. Thank you very much for this comment. This is an overlook on our side, we have now added the description of the connectivity procedure to the methods section (methods section, page 8):
“In order to construct a connected kNNG we adopt a greedy procedure. First, we build a standard kNNG for a given k. Then, for each pair of disconnected components (if any) we find the edge with the minimum length that would connect these two components. Then, we connect the two components that can be linked using the smallest edge. We repeat this process until the kNNG has only one connected component.”
- **“The authors gave no response for the comparison with scVI”**.
We are afraid we had misunderstood your comment. You suggested to use scVI in the comments to our “Interpolation” chapter, which we removed in the new version. scVI is a very good method supporting a wide range of tasks, but it is never used for dimensionality reduction to 2 dimensions. The default parameter proposed by the authors is to use it to reduce the dataset to a minimum of 10 dimensions. But 10 dimensions are not suitable for visualization, therefore authors use UMAP for this task (a method we already compare with). Alternatively, the 10-dimensional product of scVI could be used with Poincaré maps as a pre-processing step. This is a very important remark and we add it to the “Data preprocessing” paragraph in methods (page 8):
“Poincaré maps, like any other nearest-neighbors manifold learning method (e.g. UMAP or tSNE), can suffer from the curse of dimensionality. Therefore, when the dimensionality of data exceeds 100 dimensions, preprocessing to its 50-100 principal components is a widely used strategy for this type of methods. Alternatively, methods such as scVI \cite{lopez2018deep} were demonstrated to offer effective preprocessing for scRNAseq datasets, when used together with methods such as UMAP. Likewise, scVI components can be used as a preprocessing step to Poincaré maps.”
- **“scSphere”** As you know, ‘scSphere’ is a follow up work on our proposal. Regarding comparison to Ding and Regev (2019): We'd like to point out that the scSpheres manuscript was released on bioarxiv long after our manuscript (Poincaré maps v1 was posted on July 02, 2019, scSpheres v1 on November 25, 2019) and even after we first submitted this manuscript for review (July 29, 2019). In fact, scSpheres has been developed with knowledge of our work as Ding and Regev (2019) cite our manuscript. Moreover, scSpheres is not peer-reviewed yet and has a different focus than our work (batch-effects correction and generative modeling). For these reasons, we believe that a

comparison to this work is not adequate. However, since this work is related, we propose the following citation in the Introduction:

“Recently, Ding and Regev\cite{ding2019deep} proposed an interesting follow-up on our work with a focus on eliminating the batch-correction and addressing visual crowding issues of conventional generative modeling approaches via hyperbolic embeddings. This further illustrates the benefits of hyperbolic geometry for analyzing single-cell data as proposed in this work.”

Detailed response for Reviewer #2 for iteration 1:

1. I noticed the hyperbolic embedding method has already been published in the author’s another paper. To adapt this embedding method to scRNA-seq datasets,

(a) Are only kNN graph construction and global proximities to measure the distance between cells key steps for the adaptation? If so, it will be better to discuss more about the importance of these steps.

This is now discussed in Introduction: “Fourth, while we follow Nickel et al.20 to leverage the Poincaré disk as an embedding space, we are first to employ pairwise distances obtained from a nearest-neighbor graph as a learning signal to construct hyperbolic embeddings for the discovery of complex hierarchies in data.”

(b) Some guidelines for the application on different datasets should be discussed, such as different scRNA-seq protocols (such as the 10x Genomics), and as the authors claimed in the Introduction section, the CYTOF data. For example, how should the data be transferred to the embedding algorithm?

This is now discussed in Methods section: “Data preprocessing” (page 8)

(c) And how about the time cost with respect to datasets with large number of cells?

This is now discussed in Methods section: “Computational complexity and time” (page 10)

2. Construction of kNN graph. I think this is a critical step for the success of the proposed method as in some other scRNA-seq algorithms. However, the authors fail to make their operations clear enough.

(a) How about the robustness with respect to the choice of k and σ ?

We added a study on robustness to the main text and to the Supplementary Note 2.

(b) Why and how does the author enforce the connectivity of the kNN graph? Is it for the definition of RFA? If so, is it reasonable for datasets with obviously separable cell types?

We added the discussion to the main text procedure to the supplementary.

(c) The distance used to discover the nearest neighbors is also ambiguous (the author claims the use of PCA in some datasets in the Supplementary but not in the Methods section).

We added detailed discussion about usage of PCA, scVI or simple normalization depending on the data.

(d) In addition, kNN graphs constructed are usually noisy, as described in other algorithms (such as Stuart, T., et al. Cell, 2019; Bendall, S.C., et al. Cell, 2014; Manu Setty, et al. Nat Biotechnology, 2016). Does this have effects on the performance?

This doesn't have a significant effect on the performance, given that you trust your similarity measure. We addressed the issue in Supplementary Note 2 (specifically Supp Figure 3) by showing the performance measure with respect to random seed.

3. As far as I know, there may be a lot of models of the hyperbolic space, including hyperboloid, Klein, Lorentz, etc. Can the author discuss some concerns for their selection of Poincaré disk?

We chose Poincaré disk, because it is easier to interpret. Otherwise, the choice doesn't really matter because we can map between all hyperbolic models without distortion.

4. For the unseen cell prediction task, are there some requirements for the plausible prediction? For example, what kinds of properties of the unseen cell types should have for their prediction?

This section was removed.

5. Benchmark. I noticed all real datasets tested are published before 2018. Can the author test the method on some 'fashionable' datasets? **C.elegans added.** Can the author compare the method with some recently published algorithm, for example, scVI, which is also based on the neural network, as the author's prediction strategy?

We added a wide comparison with recently published methods, such as SAUCIE, PHATE and SIMLR. scVI is not used as a standalone embedding to just 2 dimensions, but is used for at dimensionality reduction to at least 10 dimensions with a consequent application of UMAP on it. Therefore, it could be also used as pre-processing step for Poincaré maps. Running scVI just to 2 dimensions would not be fair to the method (to additionally demonstrate it, we attached example of scVI on a relatively simple dataset of Paul et. al). However, we acknowledge that scVI is a good method for dimensionality reduction from several thousands genes to 10-20 scVI dimensions and could potentially replace PCA step.

6. I suggest the 'Introduction' and 'Discussion' parts should be re-organized to clarify the significance and include some in-depth discussion.

We have rewrote these sections to clarify the significance and include more in-depth discussions.

Additionally, to eliminate any concern regarding using scVI for dimensionality reduction to only 2 dimensions on Paul dataset (convergence achieved, parameters recommended by the authors):

REVIEWERS' COMMENTS:

Reviewer #1 (Remarks to the Author):

The authors have addressed all my concerns.

Reviewer #2 (Remarks to the Author):

The authors gave sufficient responses to my questions and the manuscript has been greatly improved. I have no further comments.

REVIEWERS' COMMENTS:

Reviewer #1 (Remarks to the Author):

The authors have addressed all my concerns.

Reviewer #2 (Remarks to the Author):

The authors gave sufficient responses to my questions and the manuscript has been greatly improved. I have no further comments.

The reviewers didn't raise any additional comments.